# Addressing biodiversity knowledge shortfalls in New World Helicopsychidae (Insecta, Trichoptera): Potential distribution, environmental gradients, and identification of conservation and research priority areas

Rafael Pereira [1,2]ᴑ*, Adolfo Calor[2]ᴑ

**1** Laboratório de Organismos Aquáticos – LOA, Programa de Pós-Graduação em Sistemas Aquáticos Tropicais, Universidade Estadual de Santa Cruz – (PPGSAT – UESC), Ilhéus, Bahia, Brazil, **2** Laboratório de Entomologia Aquática – LEAq, Instituto de Biologia, Programa de Pós-Graduação em Biodiversidade e Evolução, Universidade Federal da Bahia (PPGBioEvo, UFBA), Salvador, Bahia, Brazil

ᴑ These authors contributed equally to this work.
* rafaelpsilvape@gmail.com

## Abstract

This study provides an integrative analysis of Helicopsychidae biodiversity in the New World (NW), examining their distribution patterns, environmental gradients, biodiversity hotspots, and biodiversity knowledge shortfalls. We estimated species richness, delineated potential distribution, and identified areas of priority for research/conservation efforts. Our estimates indicate that up to 75% of NW Helicopsychidae species remain undescribed, with a notable Linnean shortfall at the subgenus level: approximately 50% of *H.* (*Cochliopsyche*) species and up to 77% of *H.* (*Feropsyche*) species are yet to be described. Knowledge of semaphoronts is also limited, with immature stages documented for only 15% of *H.* (*Feropsyche*) species and 6% of *H.* (*Cochliopsyche*). Distributional records are concentrated in the Brazilian subregion (between 0°–24°S), and most species have a short environmental distribution gradient. Based on currently known, priority conservation areas are in the Antilles and the Tropical Forest. For *H.* (*Feropsyche*), priority areas include low-altitude and coastal regions with low-order streams, whereas *H.* (*Cochliopsyche*) conservation priorities lie in the large river basins, especially in the Amazon and Atlantic Forests. Future research efforts should focus on *H.* (*Feropsyche*) in the Chaco and high-altitude areas of the Atlantic Forest, Cerrado, and Caatinga domains, as well as on *H.* (*Cochliopsyche*) across South America's high order river (e.g., Amazon, Orinoco, Paraná, and São Francisco Rivers). Here, highlights recent advances in the taxonomy and distribution cataloguing of NW Helicopsychidae, despite significant progress, marjory of species remain undescribed, emphasizing the need for continued research. Although well-sampled regions like the Antilles, south Central America, Amazon coastal areas, and Central Atlantic Forest Ecological Corridor should be prioritized for conservation,

**Data availability statement:** All relevant data are within the manuscript and its Supporting Information files.

**Funding:** We thank the Instituto Chico Mendes de Conservação da Biodiversidade (ICMBio) for collecting permits. ARC acknowledge the Conselho Nacional de Desenvolvimento Científico e Tecnológico (CNPq) (under Grants 303623/2015-2). RP also thanks to the Coordenação de Aperfeiçoamento de Pessoal de Nível Superior (CAPES, finance code 001, PDS-CAPES-88882.453922/2019-01, and PRAPG-CAPES-88887.986811/2024-00) for the doctoral and post-doctoral fellowship. We thank the Conselho Nacional de Desenvolvimento Científico e Tecnológico (CNPq), Programa de Apoio à Pós-Graduação (PROAP-CAPES) for financial support, and this study was financed in part by the CAPES – Finance Code 001 (PPG Biodiversidade e Evolução). We would like to thank Dra. Neusa Hamada, Dra. Ana Pes and Dr. Gleison Desidério for providing us with data from LACIA-INPA. The funders had no role in study design, data collection and analysis, decision to publish, or preparation of the manuscript.

**Competing interests:** There is no conflict of interest.

vast areas such as the Amazon lowlands, northeast and southwest Atlantic Forest, and transition areas of Atlantic Forest and Dry Diagonal remain underexplored. To the best of our knowledge, this study represents one of first empirical analysis of environmental gradients for Trichoptera species. It establishes a foundation for understanding biogeographic patterns, environmental gradients, and the identification of biodiversity hotspots and potential distribution areas for Helicopsychidae in the New World. This work will guide future research and conservation efforts for Helicopsychidae and other Trichoptera groups in the region.

## Introduction

The distributional patterns generally used the distribution of known species as a proxy [1], and, consequently, they are strongly related to biased collection efforts [2]. There are huge gaps in biodiversity knowledge related to areas neglected in terms of collection effort, fauna inventories, taxonomists, and investment in research, especially in developing countries [3,4]. These factors cause collection bias and directly affect biodiversity knowledge [5–7].

In this context, some authors have defined the different knowledge gaps, or the Biodiversity Knowledge Shortfalls (BKS), in various categories such as gap of species knowledge is the Linnaean shortfall, the unknown species geographic distribution is the Wallacean shortfall, the gap of knowledge about semaphoronts of species Haeckelian shortfall, the gap of knowledge about abiotic tolerance of species Hutchinsonian shortfall and unknown of evolution of groups is the Darwinian shortfall [8,9]. Among the tools available to face the biodiversity knowledge shortfalls, distribution modelling can identify areas with a high probability of potential distribution (environmental suitability) [10]. The generated models can help to prioritize resources and efforts in areas with the probability of potential distribution of species [11].

On the other hand, anthropogenic actions have caused planetary-scale changes, which have caused the sixth mass extinction [12] with critical loss of biodiversity faster than our ability to catalogue it [13]. Among the ecosystems, freshwater environments cover only 1% of Earth's land surface but comprise around 10% of all species [14]. Levels of endemism in these environments are remarkably high, and about 20% of the New World (NW – here composed for West and East Nearctic, Neotropical and Patagonian region *sensu* de Moor & Ivanov) species are threatened *sensu* IUCN [15]. Due to these characteristics, freshwater environments have the most acute biodiversity crisis among ecosystems [16]. The formulation of strategies based on large datasets is urgent to protect species, as well as the priority areas [17], and consequently reduce (or reverse) the decline of freshwater biota as much as possible [16].

Several insect orders have species with life stages occurring in freshwater ecosystems, but some orders stand out as having primarily aquatic origins (e.g., Trichoptera) [18]. Among these amphibiotic orders, the caddisflies stand out as having the greatest richness and diversity of functional trophic groups [19]. Despite recent advances in caddisfly taxonomy and phylogeny, the order is underexplored and poorly known

in several aspects [e.g., 18–22]. Knowledge about Trichoptera species, distribution ranges, and biogeographic patterns is scarce, especially in the Neotropical region (NT) *sensu* de Moor & Ivanov [20,23–26].

Studies of Trichoptera have focused almost entirely on systematics, with few studies related to biogeography and a much smaller proportion concerned with understanding which factors limit the distribution of groups [25]. However, data related to the biology, environmental and biogeography limits are essential for understanding how species will behave both in scenarios of environmental impacts (e.g., deforestation, burning) and in scenarios of climate change [27]. Especially when it comes to taxa that play a key role in the trophic dynamics of freshwater and adjacent terrestrial ecosystems, such as the Trichoptera [19], and with factors such as temperature, rainfall, and riparian vegetation directly influencing the distribution of species [28]. Despite its importance, it has suffered alarming population declines, 2/3 (68%) of the caddisfly species experiencing population declines, 63% of the species threatened, and a local/regional extinction rate of 6.8% [29].

Among the caddisflies, Helicopsychidae von Siebold, 1856 are represented by two genera, the monotypic *Rakiura* McFarlane, 1973 (endemic to New Zealand), and *Helicopsyche* von Siebold, 1856 (circum-global distribution) [30]. *Helicopsyche* comprises 298 valid species in six subgenera: *Cochliopsyche* Müller, 1885, *Feropsyche* Johanson, 1998, *Galeopsyche* Johanson, 1998, *Helicopsyche*, *Petrotrichia* Ulmer, 1910, *Saetotricha* Brauer, 1865 [31]. In the Neotropical (NT), Nearctic (NA) and Patagonian (PA) regions (*sensu* de Moor & Ivanov) have 146 valid species in two subgenera, *H.* (*Cochliopsyche*), with 17 Neotropical species, and *H.* (*Feropsyche*), with 129 species (including three fossil species from the Dominican Republic) widely distributed throughout the NW (NA with 13, NT with 125, and PA with two valid species) [31,32].

*Helicopsyche* is a genus recognized by larval cases built with sand grains, helically organized, and resembling snail shells [33]. Like most caddisflies, the immature stages live in freshwater ecosystems and, after completing the metamorphosis, they emerge as winged adults associated with riparian forests [34]. Due to its wide distribution and occurrence in different freshwater environments of the NW [35], the genus represents a good biological model for initial studies that seek to identify and define distributional patterns of amphibiotic insect groups. Furthermore, understanding how these species are distributed and grouped can provide subsidies for the identification of under-sampled areas [36].

The subgenus *Feropsyche* was revised by Johanson (2002), and subsequent studies have primarily focused on species descriptions and new distribution records [e.g., 32,37–40]. In contrast, the subgenus *Cochliopsyche* has received comparatively less attention following its revision by Johanson (2003) [41], with most contributions limited to new distribution records [e.g., 42–44] and the description of a single new species, *Helicopsyche* (*Cochliopsyche*) *nyurga* Oláh & Oláh, 2022. The taxonomic literature on *Helicopsyche* also reveals a notable bias toward male specimens [33,45].

Species description based only on male adults is a quite frequent practice in caddisfly taxonomy and accurately separates species [24], as in some other insect orders [7]. For the most caddisflies, immature stages and females are unknown (an average of <15% of the immatures are known, and in 41 Neotropical genera, the immature stages are unknown) [46], demonstrating a knowledge shortfall related to other semaphoronts. This causes a genuine problem since most adult collections use light traps, which collect more adult females than males [47]. In addition, *Helicopsyche* species are mainly reported exclusively to type localities and/or adjacent localities (like the same district) [e.g., 32]. The combination of these two factors inevitably leads to knowledge shortfalls [8].

Understanding distribution patterns, environmental gradients, and biodiversity hotspots—along with identifying Biodiversity Knowledge Shortfalls (KBS) and prioritizing research efforts—provides crucial insights for addressing Wallacean and Hutchinsonian shortfalls and informs strategies for tackling other BKS. Additionally, this information forms a foundation for developing new research proposals, conservation actions, and enhancing understanding of biogeographic patterns and the relationships between species and regions for New World Trichoptera. Within this framework, this study aimed to identify distribution patterns, establish environmental gradients, and model the distribution of Helicopsychidae subgenera in the New World.

## Materials and methods

### Study area

The New World is divided into three regions: the Nearctic, Neotropical, and Patagonian regions, with two transition zone: (i) Mexican transition zone comprise mountainous areas of central and southern Mexico and northern Central America, serves as a pathway connecting elements of the Nearctic and Neotropical fauna; and (ii) South American transition zone comprise highlands of the Andes between western Venezuela and northern Chile and central western Argentina [48]. Following these delimitations and using the distribution of Trichoptera as a proxy de Moor & Ivanov [20] divided Nearctic region into three regions: (i) West Nearctic (=Western subregion), (ii) East Nearctic (=Alleghany subregion), and (iii) Beringian "in part" (=Artic subregion) [14]. In addition to two regions covering part of Mexico, Central and South America, the Neotropical region (=Neotropical region) which includes Antillean, Brazilian, and Chacon subregions [*sensu* 48,49], and Patagonian region (=Andean or Chilean region) which includes Central Chilean, Subantarctic and Patagonian subregions [*sensu* 50]. Apart from these, all other proposals for bioregionalization for aquatic insects have been made with too large divisions [e.g., 51], making it difficult to use. Here we'll follow the gradients of de Moor & Ivanov' proposal [20] and when necessary subregions of Morrone and Escalante [48–50].

### Distributional data and richness estimates

The distribution data basis was compiled through the primary literature (description and record of the occurrence of species), the database of Global Biodiversity Information Facility (GBIF, https://www.gbif.org), and SpeciesLink (http://www.splink.org.br/). Original data were obtained from collections at the Museu de História Natural da Bahia of the Universidade Federal da Bahia (UFBA), the Laboratório de Citotaxonomia e Insetos Aquáticos of the Instituto Nacional de Pesquisas da Amazônia (LACIA-INPA), the Museum of Comparative Zoology at Harvard University (MCZ), the University of Minnesota Insect Collection (UMSP), and the National Museum of Natural History of the Smithsonian Institution (USNM). As the study was based on previously collected and curated material from these museums and institutional collections, no additional permits were required for the collection or transportation of biological specimens.

We use gazetteers and Google Maps© to register localization without coordinates; the centroid of the less comprehensive location was used. After the data compilation, two-stage filtering process was performed, (i) manual selection of the data, discarding points without coordinate information, generic data of locality (e.g., only state), or with the indeterminacy of the species (e.g., identification only to the genus level), and (ii) selection in R environment, discarding points that can generate an analysis bias (e.g., points localized at the centroid of the capital areas and/or with same coordinates or in marine areas). Species distribution map and heatmap (Kernel density) were prepared using QGIS v. 3.10.10. Distributional records of 146 species, of which 17 species are *H.* (*Cochliopsyche*) and 129 species are *H.* (*Feropsyche*) (Table 1; S1 Table) and five undescribed species of *H.* (*Cochliopsyche*) (being described in another paper) were catalogued.

For estimating the number of unknown species of *H.* (*Feropsyche*) in NA-NT-PA and *H.* (*Cochliopsyche*) in NT, the distributional records and bioregions delimited here are used, using non-parametric estimators. Estimators were calculated based on incidence data (presence-absence only), using freshwater ecoregions (Abell et al. 2008) as sampling unities, counting the occurrences in each one, with the function 'specpool' from the 'vegan' package [52] in the R environment. This function calculates two estimators of species richness: CHAO2, second-order jackknife (JACK2) (more adequate for incidence data) [53]. These non-parametric estimators are useful to estimate a potential number of unobserved species based on incidence data as those available here, and they have shown better performance than model-based or asymptotic estimators [54,55].

### Environmental gradients

To obtain the standards of distribution and distribution range regarding temperature, precipitation, elevation, and characteristics of freshwater environments, a dataset was produced consisting of distribution data, biogeoclimatic data (i.e.,

**Table 1.** Species of *Helicopsyche* and information of known semaphoronts, distribution, and collections with deposited material.

| Species | KS | Museum | Distribution |
|---|---|---|---|
| *H. (C.) amazona* Johanson, 2003 | ♂ | **USNM**, NRM | BRA |
| *H. (C.) amica* Johanson, 2003 | ♂, ♀* | MVC, USNM, **MCZ**, ISNB | BRA, GUY, VEN |
| *H. (C.) blahniki* Johanson, 2003 | ♂, ♀* | **UMSP**, CIUC, FMNH, MVC, USNM NRM | BRA, COL, ECU, GUY, PER, VEN |
| *H. (C.) brazilia* Johanson, 2003 | ♂, ♀* | **MZUSP**, USNM, MRM | BRA |
| *H. (C.) chocoensis* Johanson, 2003 | ♂, ♀* | **USNM** | BRA, COL |
| *H. (C.) clara* (Ulmer, 1905) | ♂, ♀ | **MCZ**, MZUSP, UMSP, USNM | ARG, BRA, ECU |
| *H. (C.) holzenthali* Johanson, 2003 | ♂, ♀* | **UMSP**, USMN | VEN |
| *H. (C.) lobata* (Flint, 1983) | ♂ | **USNM**, MCZ, MZUSP, UMSP | ARG, BRA, PER |
| *H. (C.) nyurga* Oláh & Oláh, 2022 | ♂ | **OPC** | ECU |
| *H. (C.) napoa* Johanson, 2003 | ♂, ♀* | **USNM** | ECU |
| *H. (C.) ocosingua* Johanson, 2003 | ♂, ♀* | **INHS**, NRM | BRA, MEX |
| *H. (C.) opalescens* (Flint, 1972) | ♂ | **USNM**, FNMH, MZUSP, UMSP | ARG, BRA, ECU, GUY, PAR, PER, SUR, URU, VEN |
| *H. (C.) pandeirosa* Johanson, 2003 | ♂, ♀* | **MZUSP**, UMSP, NRM, USNM | BRA |
| *H. (C.) paraguaiensis* Johanson, 2003 | ♂ | **USNM** | PAR |
| *H. (C.) puyoa* Johanson, 2003 | ♂, ♀* | **USNM**, UMSP | BRA, ECU |
| *H. (C.) vazquezae* (Flint, 1986) | ♂, L, P, C | **USNM**, INHS, UMSP | BOL, COR, ECU, MEX, VEN |
| *H. (C.) xinguensis* Johanson, 2003 | ♂, ♀* | **MZUSP**, USNM, UMSP | BRA |
| *H. (F.) alajuela* Johanson & Holzenthal, 2010 | ♂ | **NMNH**; COZEM | CRI; PAN |
| *H. (F.) altercoma* Botosaneanu & Flint, 1991 | ♂♀ | **NMNH**; CMNH; FSCA; ZMUA | DOM |
| *H. (F.) angeloi* Holzenthal, Blahnik & Calor, 2016 | ♂♀* | **MZUSP**; UFBA; UMSP | BRA |
| *H. (F.) angulata* Flint, 1981 | ♂♀ | **USNM**; UMSP; NRM | COL; ECU; VEN |
| *H. (F.) apicauda* Flint, 1968 | all | **USNM**; NMNH | DMA; GUA |
| *H. (F.) auroa* Johanson & Holzenthal, 2004 | ♂♀* | **UMSP**; NMNH | VEN |
| *H. (F.) bendego* Dumas & Nessimian, 2019 | ♂♀* | **DZRJ** | BRA |
| *H. (F.) blancasi* Schmid, 1958 | ♂♀* | **NMNH** | PER |
| *H. (F.) blantoni* Johanson & Malm, 2006 | ♂ | **NMNH**; NRM | PAN |
| *H. (F.) borealis* (Hagen, 1861) | all | **MCZ**; UMSP; NMNH; TAMU; USNM | CAN; CRI; GTM; HND; MEX; NIC; PAN; USA |
| *H. (F.) braziliensis* (Swainson, 1840) | C | – | BRA |
| *H. (F.) breviterga* Flint, 1991 | ♂♀* | **UMSP**; COZEM; UMSP; NRM | COL; PAN; VEN |
| *H. (F.) caligata* Flint, 1967 | ♂ | **NMNH** | CHI |
| *H. (F.) camuriensis* Johanson & Holzenthal, 2004 | ♂♀* | **UMSP** | VEN |
| *H. (F.) carajas* Gama Neto, Ribeiro & Passos, 2019 | ♂ | **MPEG** | BRA |
| *H. (F.) catoles* Souza, Gomes & Calor, 2017 | ♂♀* | **MZUSP**; UFBA; UFRJ | BRA |
| *H. (F.) centrocubana* Botosaneanu & Flint, 1991 | ♂L | **ZMUA** | CUB |
| *H. (F.) chilensis* Flint, 1983 | ♂♀* | **NMNH** | CHI |
| *H. (F.) chiriquensis* Johanson & Malm, 2006 | ♂♀* | **NMNH**; UMSP; INBIO | CRI; PAN |
| *H. (F.) cipoensis* Johanson & Malm, 2006 | ♂ | **NMNH** | BRA |
| *H. (F.) circulata* Johanson & Holzenthal, 2004 | ♂ | **UMSP** | VEN |
| *H. (F.) cochleara* Johanson, 1999 | ♂ | **NMNH** | ECU |
| *H. (F.) colombiensis* von Siebold, 1856 | C | – | COL; VEN |
| *H. (F.) comosa* Kingsolver, 1964 | ♂♀ | **INHS**; ZMUA; NMNH; MCZ | CUB |
| *H. (F.) cotopaxi* Botosaneanu & Flint, 1982 | ♂♀*L*P*C | **USNM**; ZMUA | ECU |

*(Continued)*

| Species | KS | Museum | Distribution |
|---|---|---|---|
| *H. (F.) cubana* Kingsolver, 1964 | all* | **INHS**; NHMJ; ZMUA | CUB; JAM |
| *H. (F.) curvipalpia* Johanson & Malm, 2006 | ♂♀* | **INHS**; NRM | MEX |
| *H. (F.) dampfi* Ross, 1956 | ♂♀*P* | **INHS**; CNHM; NMNH; UMSP; INBIO; MEL | CRI; GTM; MEX; NIC |
| *H. (F.) daome* Dumas & Nessimian, 2019 | ♂♀* | **DZRJ** | BRA |
| *H. (F.) diamantina* Pereira & Calor, 2024 | ♂ | **MZUSP; UFBA** | BRA |
| *H. (F.) dinoprata* Dumas & Nessimian, 2019 | ♂♀* | **DZRJ**; MZUSP | BRA |
| *H. (F.) disjuncta* Johanson & Holzenthal, 2004 | ♂ | **NMNH** | VEN |
| *H. (F.) dominicana* Botosaneanu & Flint, 1991 | ♂♀ | **USNM; CMNH; CNHM; FSCA; NMNH; ZMUA** | DOM |
| *H. (F.) dorsocurvata* Johanson & Holzenthal, 2010 | ♂ | **UMSP** | CRI |
| † *H. (F.) electra* Johanson & Wichard, 1996 | ♂ | Collection Wichard | DOM |
| *H. (F.) extensa* Ross, 1956 | ♂♀* | **INHS**; UMSP | PER; VEN |
| *H. (F.) falcigona* Botosaneanu & Flint, 1991 | ♂♀*L | **ZMUA**; USNM; MCZ | CUB |
| *H. (F.) fistulata* Flint, 1991 | ♂♀* | **USNM** | COL; VEN |
| *H. (F.) flinti* Johanson, 1999 | ♂ | **BMNH** | BRA |
| *H. (F.) fridae* Johanson, 2003 | ♂♀* | **NMNH**; UCD | PAN |
| *H. (F.) golfitoensis* Johanson & Holzenthal, 2010 | ♂ | **NMNH** | CRI |
| *H. (F.) granpiedrana* Botosaneanu & Sykora, 1973 | ♂ | **ZMUA** | CUB |
| *H. (F.) grenadensis* Flint & Sykora, 1993 | ♂♀* | **FSCA**; NMNH; UMSP; NRM | GRE; VEN |
| *H. (F.) guadeloupensis* Malicky, 1980 | all | **CM**; ZMUA; MHNH; CMNH; MNNM; MNHM | DMA; GUA; LCA; MTQ |
| *H. (F.) guara* Holzenthal, Blahnik & Calor, 2016 | ♂♀* | **MZUSP**; UMSP; UFBA | BRA |
| *H. (F.) guariru* Vilarino & Calor, 2017 | ♂ | **MZUSP**; UMSP; UFBA | BRA |
| *H. (F.) hageni* Banks, 1938 | all | **MCZ**; ZMUA | CUB; DOM |
| *H. (F.) haitiensis* Banks, 1938 | ♂ | **MCZ** | HTI |
| *H. (F.) helicoidella* (Vallot, 1855) | C | – | BRA |
| *H. (F.) incisa* Ross, 1956 | ♂♀* | **INHS**; UMSP; INBIO; NMNH; NRM | CRI; MEX; NIC; PAN |
| *H. (F.) imperial* Silva-Pereira, Desidério, Pereira, Hamada, 2024 | ♂ | **INPA, UFBA** | BRA |
| *H. (F.) johansoni* Moreno, Desidério, Pes & Hamada, 2023 | ♂ | **INPA; DZRJ; MNRJ; UFBA** | BRA |
| *H. (F.) inflata* Gama Neto, Ribeiro & Passos, 2019 | ♂ | **MPEG** | BRA |
| *H. (F.) kalaom* Botosaneanu, 1996 | ♂♀* | **ZMUA**; FSCA; NMNH; CMNH | DOM |
| *H. (F.) kingstona* Johanson, 2003 | ♂ | **UCD** | JAM |
| *H. (F.) krenak* Bonfá-Neto, Vilarino & Salles, 2023 | ♂ | **UFVB** | BRA |
| *H. (F.) lambda* Flint, 1983 | ♂ | **NMNH** | ARG |
| *H. (F.) laneblina* Johanson & Holzenthal, 2004 | ♂ | **NMNH** | VEN |
| *H. (F.) lara* Johanson & Holzenthal, 2004 | ♂♀* | **UMSP**; IZAM; NRM | VEN |
| *H. (F.) lazzariae* Holzenthal, Blahnik & Calor, 2016 | ♂ | **MZUSP** | BRA |
| *H. (F.) lewalleni* Denning & Blickle, 1979 | ♂♀ | **CAS**; INBIO; UMSP | CRI; ELS |
| *H. (F.) limnella* Ross, 1937 | ♂ | **INHS** | **USA** |
| *H. (F.) linabena* Johanson & Holzenthal, 2004 | ♂♀* | **NMNH** | VEN |
| *H. (F.) linguata* Johanson & Malm, 2006 | ♂ | **NMNH** | PAN |
| *H. (F.) lutea* (Hagen, 1961) | ♀ | **MCZ** | DOM |
| *H. (F.) luziae* 17 | ♂♀* | **DZRJ** | BRA |

*(Continued)*

| Species | KS | Museum | Distribution |
|---|---|---|---|
| H. (F.) maculisternum Botosaneanu, 1993 | ♂♀* | **ZMUA** | VEN; TRI |
| H. (F.) manaos Moreno, Desidério, Pes & Hamada, 2023 | ♂ | **NMNH**; ZMUA; CMNH; FSCA | DOM |
| H. (F.) melanochaeta Flint & Sykora, 2004 | ♂♀* | **INPA; DZRJ; MNRJ; UFBA** | BRA |
| H. (F.) merida Botosaneanu & Flint, 1982 | ♂♀*L*P*C | **NMNH**; ZMUA; UMSP | VEN |
| H. (F.) mexicana Banks, 1901 | ♂♀ | **MCZ;** INHS; NMNH; USNM; CAS; OSU; UCR | CRI; MEX; USA |
| H. (F.) minima von Siebold, 1856 | all | **USNM**; ZMUA | NIC; PRI |
| H. (F.) minuscula Martynov, 1912 | ♀ | **PAN** | PER |
| H. (F.) molesta Botosaneanu, 1998 | ♂ | **ZMUA** | JAM |
| H. (F.) mateusi Pereira & Calor, 2024 | ♂ | **MZUSP; UFBA** | BRA |
| H. (F.) miltonsantosi Pereira & Calor, 2024 | ♂ | **MZUSP; UFBA** | BRA |
| H. (F.) monda Flint, 1983 | ♂ | **USNM**; DZRJ; NMNH | ARG; BRA; PRY; VEN |
| H. (F.) montana Felber, 1912 | LPC | **NMB** | MEX |
| H. (F.) muelleri Banks, 1920 | ♂♀*LPC | **MCZ**; IRSNB | ARG; BRA; PER |
| H. (F.) neblinensis Johanson & Holzenthal, 2004 | ♂♀* | **NMNH**; IZAM; NRM | VEN |
| H. (F.) nigrisensilla Botosaneanu & Flint. 1991 | ♂♀ | **USNM; ZMUA** | DOM |
| H. (F.) obscura Rueda Martín & Isa Miranda, 2015 | ♂LPC | **IBN** | ARG |
| H. (F.) occidentale Botosaneanu & Flint, 1991 | ♂L | **USNM; ZMUA** | CUB |
| H. (F.) ochthephila Flint, 1968 | all | **NMNH** | JAM |
| H. (F.) paprockii Johanson & Malm, 2006 | ♂ | **NMNH** | BRA |
| H. (F.) parahageni Flint & Sykora, 2004 | ♂♀ | **NMNH;** CMNH; FSCA | DOM |
| H. (F.) paralimnella Hamilton, 1989 | ♂ | **CUEC** | USA |
| H. (F.) paucispina Botosaneanu & Flint, 1991 | ♂ | **ZMUA** | CUB |
| H. (F.) perija Johanson & Holzenthal, 2004 | ♂ | **UMSP** | VEN |
| H. (F.) peruana Banks, 1920 | ♂ | **MCZ** | PER |
| H. (F.) paulofreirei Pereira & Calor, 2024 | ♂ | **MZUSP; UFBA** | BRA |
| H. (F.) petri Dumas & Nessimian, 2019 | ♂ | **DZRJ; MZUSP** | BRA |
| H. (F.) pietia Denning, 1964 | ♂♀ | **CAS;** INHS; NRM | MEX; USA |
| H. (F.) piroa Ross, 1944 | ♂♀* | **INHS;** TAMU; USNM | CRI; MEX; NIC; USA |
| H. (F.) planata Ross, 1956 | ♂ | **INHS;** CNIN; CUEC | NIC; MEX |
| H. (F.) planorboides Machado, 1957 | all | **DZRJ** | BRA |
| H. (F.) poliochaeta Flint & Sykora, 2004 | ♂♀ | **NMNH;** FSCA | DOM |
| H. (F.) propinqua Botosaneanu & Flint, 1991 | ♂ | **NMNH** | PRI |
| H. (F.) quadrosa Ross, 1956 | ♂ | **INHS** | MEX |
| H. (F.) ralphi Cavalcante-Silva, Pereira & Calor, 2022 | all | **MZUSP;** INPA; UFBA; UFRJ | BRA |
| H. (F.) ramosi Flint, 1964 | all | **NMNH;** ZMUA | PRI |
| H. (F.) rentzi Denning & Blickle, 1979 | ♂♀ | **CAS;** INBIO; UMSP; USNM | CRI |
| H. (F.) sanblasensis Johanson & Malm, 2006 | ♂ | **NMNH** | PAN |
| H. (F.) scalaris Hagen, 1864 | C | – | VEN |
| † H. (F.) electra Johanson & Wichard, 1996 | ♂ | **Collection Wichard** | DOM |
| H. (F.) selanderi Ross, 1956 | ♂♀* | **INHS;** NMNH; UMSP | CRI; MEX; VEN |
| H. (F.) septifera Flint & Sykora, 2004 | ♂♀ | **NMNH;** CMNH; ZMUA | DOM |
| H. (F.) shaamunensu Dumas & Nessimian, 2019 | ♂ | **DZRJ** | BRA |
| H. (F.) sigillata Botosaneanu & Flint, 1991 | ♂♀ | **NMNH; ZMUA** | CUB |

*(Continued)*

**Table 1.** (Continued)

| Species | KS | Museum | Distribution |
|---|---|---|---|
| *H. (F.) singulare* Botosaneanu & Flint, 1991 | ♂ | **NMNH**; USNM; ZMUA | PRI |
| *H. (F.) sinuata* Denning & Blickle, 1979 | ♂ | **UCD**; NMNH | MEX; USA |
| *H. (F.) succincta* Johanson & Holzenthal, 2004 | ♂ | **NMNH**; UFBA | BRA; VEN |
| *H. (F.) sucrensis* Johanson & Holzenthal, 2004 | ♂ | **UMSP** | VEN |
| *H. (F.) tachira* Johanson & Holzenthal, 2004 | ♂ | **UMSP** | VEN |
| *H. (F.) tapadas* Denning, 1966 | ♂ | **CAS**; UFBA | BRA; VEN |
| *H. (F.) temora* Denning & Blickle, 1979 | ♂ | **UCD** | MEX |
| *H. (F.) thelidomus* Hagen, 1864 | C | – | VEN |
| *H. (F.) timbira* Silva, Santos & Nessimian, 2014 | ♂♀* | **DZRJ; MNRJ; INPA** | BRA |
| *H. (F.) truncata* Ross, 1956 | ♂ | **INHS;** UMSP; USNM | CRI; MEX; PAN |
| *H. (F.) turbida* Navás, 1923 | all | **MZBS**; NMNH; IBN; USNM | ARG |
| *H. (F.) tuxtlensis* Bueno-Soria, 1983 | ♂♀* | **IBUNAM**; UCD; USNM | GTM; MEX; PAN |
| *H. (F.) umbonata* Hagen, 1864 | all | **USNM**; MCZ; AMNH; USNM | JAM |
| *H. (F.) valligera* Flint, 1983 | ♂♀* | **NMNH**; USNM | ARG; BRA |
| *H. (F.) venezuelensis* Johanson & Holzenthal, 2004 | ♂ | **UMSP** | VEN |
| *H. (F.) vergelana* Ross, 1956 | all | **INHS**; NMNH; NRM; UCD; UFPE; UMSP; USNM | BLZ; BRA; COL; CRI; ECU; GTM; GUY; MEX; NIC; PAN; PER; SUR; TRI; VEM |
| *H. (F.) villegasi* Denning & Blickle, 1979 | ♂ | **UCD** | MEX |
| † *H. (F.) voigti* Johanson & Wichard, 1996 | ♂ | **Collection Wichard** | DOM |
| *H. (F.) woldai* Johanson, 2003 | ♂ | **NMNH**; UCD | PAN |
| *H. (F.) woytkowskii* Ross, 1956 | ♂♀* | **INHS**; NRM; UMSP | PAN; PER; VEN |

KS = Known semaphoronts; ♂ = Male; ♀ = Female; L = Larvae; P = Pupa; Collections that house type material in bold; *semaphoronts present in material examined of the publications, but not formally described.

temperature, precipitation, elevation) obtained through WorldClim version 2.1 (https://www.worldclim.org/data/world-clim21.html), and data of freshwater ecosystems (i.e., Sthaler order, flow velocity, discharge, water body size) available in HydroRIVERS and HydroLAKES version 1.0 (https://www.hydrosheds.org). Based on this, distribution value plots and boxplots were generated using the 'ggplot' function of the ggplot2 package, a correlation analysis between the variables was performed with the Spearman method using the 'correlate' function of the 'corrr' package (https://cran.r-project.org/package=corrr) and PERMANOVA analysis with Jaccard similarity index was performed to assess the similarity between subgenus both performed in R environment.

## Distribution modelling

Environmental data were obtained from monthly climate data for minimum, mean, and maximum temperature, precipitation, solar radiation, wind speed, water vapor pressure, and for total precipitation, 19 "bioclimatic" variables, and elevation on a scale of 5 arc minutes, all environmental variables obtained in WorldClim version 2.1 (https://www.worldclim.org/data/worldclim21.html). The resolutions of these environmental variables were kept, avoiding the loss of information or the impossibility of modelling. After obtaining the data, a correlation analysis between the variables was performed with the Spearman method using the 'correlate' function of the 'corrr' package (https://cran.r-project.org/package=corrr) in the R environment. This correlation analysis permitted to selection of uncorrelated variables and consequently to avoid over-weighing in the models. Variables with correlation values greater than 80% were considered correlated.

Four correlative modelling algorithms were used: Bioclim [56], Domain [57], Generalized Linear Model (GLM) [58], and the Vector Support Machine (SVM) [59]. As absence data is not predicted or lacks accuracy given the bias/scarce sampling scenario, presence/pseudo-absence models will be used. We randomly generated pseudo-absence points (1:1 ratio for the occurrence points) [60], through the 'randomPoints' function of the 'dismo' package (https://cran.r-project.org/package=dismo) in the R environment. Data partitioning was randomly performed in 70% for training and 30% for testing the models. The repeatability of the models (200 times) was used to increase the robustness of the result. After the models were evaluated using the Area Under the Curve (AUC) [61] method, which only models with values of AUC > 80% were used for the construction of suitability maps. Finally, we used the default limits of presence and absence for the construction of the suitability maps.

## Results

### Biodiversity knowledge shortfalls

In the New World, the *Helicopsyche* species richness remains largely underestimated, with estimates suggesting up to 571 species (error = 142), indicating that approximately 75% of the species remain undescribed (Table 2). The subgenus *H.* (*Cochliopsyche*) comprises 17 valid species, all distributed in the Neotropics, particularly in the Brazilian subregion (Fig 1B). Estimates project 36 species (error = 16), with 53% remaining undescribed (Table 2). The first species was described by Ulmer in 1905, followed by significant contributions from Flint [62–64]. Johanson's revision added 12 species, and after a 20-year gap, Oláh & Oláh [65] described a new species from Ecuador (Fig 2C). The subgenus *H.* (*Feropsyche*) includes 129 valid species, widely distributed across tropical regions of the Nearctic, Neotropical and Patagonian regions, with the highest richness in the Brazilian subregion (125 species) (Fig 1G). Estimates suggest 553 species (error = 152), with 77% remaining undescribed (Table 2). In the past 20 years, about 40% of *H.* (*Feropsyche*) species were described, averaging five species annually, compared to two species per year from 1840 to 1999 (Fig 2D).

Approximately 79% of *Helicopsyche* species have restricted distributional records (< 10 records), with around 70% occurring only in type localities or adjacent areas, such as the same stream (Fig 2A and B; S1 Table). The species with the widest distribution are *H.* (*Cochliopsyche*) *opalescens* Flint, 1972, with 41 records across 147.3 km², and *H.* (*Feropsyche*) *borealis* (Hagen, 1861), with 48 records across 211.03 km² (Fig 2A and B). In *H.* (*Cochliopsyche*), 58% of species have restricted records, with 47% limited to type localities or adjacent areas (Fig 2A; S1 Table). In contrast, *H.* (*Feropsyche*) shows a higher percentage of species with restricted records, reaching 88%, and 72% of these species are confined to type localities or adjacent areas (Fig 2B; S1 Table).

Immature stages are documented for 20 valid species (15%), and female adults are described for 30 valid species (24%). In *H.* (*Cochliopsyche*), adult males of all species are described. However, immature stages are known for only one

**Table 2. Summary of the estimation of species richness in the new world based on incidence data with the CHAO2, second order jackknife (JACK2) estimators.**

|  | Chao2 | Jacknife2 | Average | Estimation of unknown species |
|---|---|---|---|---|
| *Helicopsyche (Cochliopsyche)* |  |  |  |  |
| Species estimates | 36 | 34 | 35 | 18 |
| Error | 16 | 4 | 10 |  |
| *Helicopsyche (Feropsyche)* |  |  |  |  |
| Species estimates | 553 | 310 | 431 | 302 |
| Error | 152 | 33 | 92 |  |
| New World *Helicopsyche* |  |  |  |  |
| Species estimates | 571 | 343 | 457 | 311 |
| Error | 142 | 35 | 88 |  |

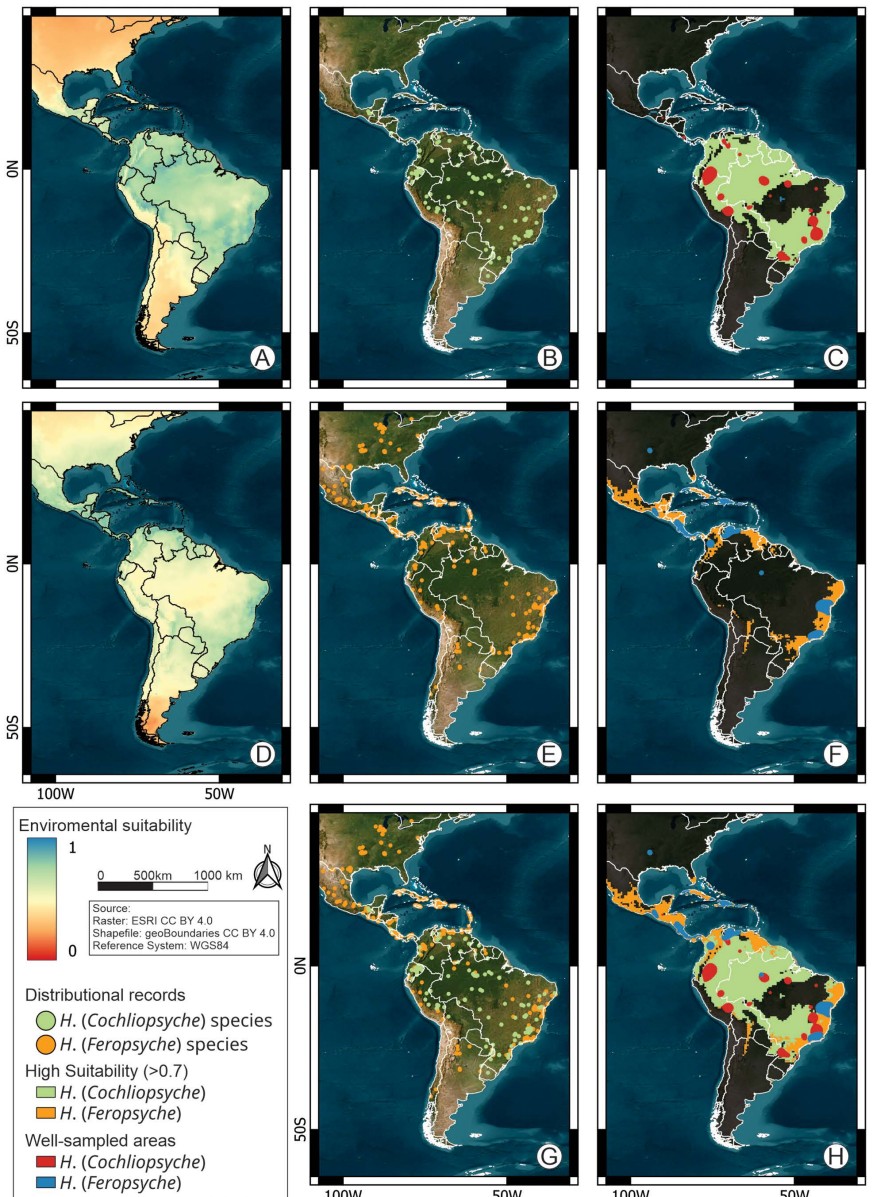

**Fig 1. Distribution and environmental suitability map for the New World Helicopsychidae species.** A. Environmental suitability map (weighted average) of *Helicopsyche* (*Cochliopsyche*); B. Distribution records of *Helicopsyche* (*Cochliopsyche*) species; C. Map of high environmental suitability (>0.7) of *Helicopsyche* (*Cochliopsyche*); D. Environmental suitability map (weighted average) of *Helicopsyche* (*Feropsyche*); E. Distribution records of *Helicopsyche* (*Feropsyche*) species; F. Map of high environmental suitability (>0.7) of *Helicopsyche* (*Feropsyche*); G. Distribution records of New World Helicopsychidae species; H. Map of high environmental suitability (>0.7) of New World Helicopsychidae. Map created in QGIS v3.40.7. Background: ESRI land cover (https://livingatlas.arcgis.com/landcover/) accessed via the QuickMapServices plugin, country boundaries: geoBoundaries (https://www.geoboundaries.org/), both under CC BY 4.0 license.

species, *H.* (*Cochliopsyche*) *vazquezae* (Flint, 1986), and females are documented for two species, *H.* (*Cochliopsyche*) *clara* (Ulmer, 1905) and *H.* (*Cochliopsyche*) *vazquezae* (Table 1). In *H.* (*Feropsyche*), males of 121 species and females of 28 species are described, including two species known solely from females: *H.* (*Feropsyche*) *lutea* (Hagen, 1861) and *H.* (*Feropsyche*) *minuscula* Martynov, 1912. Immature stages have been documented for 19 species of *H.* (*Feropsyche*).

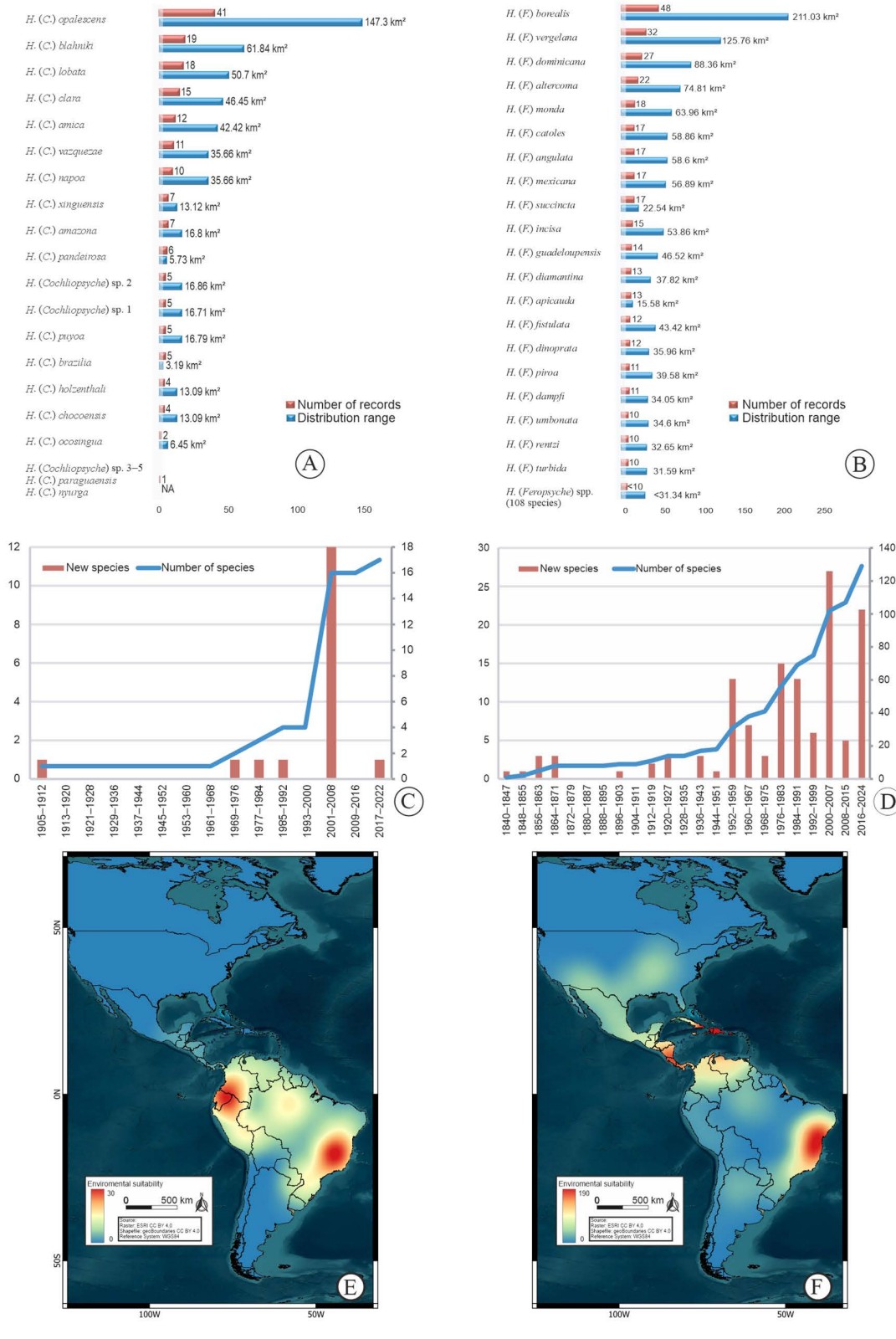

**Fig 2. Distribution range and species new species history of New World Helicopsychidae.** A. Number of records and distribution range of *Helicopsyche* (*Cochliopsyche*) species. B. Number of records and distribution range of *Helicopsyche* (*Feropsyche*) species. C. History of new and number of *Helicopsyche* (*Cochliopsyche*) species over time. D. History of new and number of *Helicopsyche* (*Feropsyche*) species over time. E. Heatmap of distribution for *Helicopsyche* (*Cochliopsyche*). F. Heatmap of distribution for *Helicopsyche* (*Feropsyche*). Map created in QGIS v3.40.7. Background: ESRI land cover (https://livingatlas.arcgis.com/landcover/) accessed via the QuickMapServices plugin, country boundaries: geoBoundaries (https://www.geoboundaries.org/), both under CC BY 4.0 license.

Of these, seven species are known exclusively from their larval cases: *H.* (*Feropsyche*) *braziliensis* (Swainson, 1840), *H.* (*Feropsyche*) *colombiensis* von Siebold, 1956, *H.* (*Feropsyche*) *helicoidella* (Vallot, 1855), *H.* (*Feropsyche*) *minima* von Siebold, 1956, *H.* (*Feropsyche*) *scalaris* Hagen, 1864, *H.* (*Feropsyche*) *thelidomus* Hagen, 1864, and *H.* (*Feropsyche*) *umbonata* Hagen, 1864 (Table 1).

Regarding environmental gradients, *Helicopsyche* species occur along an elevation gradient from 0 to 5042 m above sea level (a.s.l.). With *H.* (*Cochliopsyche*) species range from 3 to 2457 m, with 80% found below 600 m (Fig 3A). In contrast, *H.* (*Feropsyche*) species range from 0 to 5042 m, with 45% of records above 600 m (Fig 3B), showing significant differences between subgenera distributions ($F_{(1)}$=35.19, *p*-value=0.001). *Helicopsyche* species are found across the 1st to 10th Strahler stream orders. The *H.* (*Cochliopsyche*) species has nearly half of its records above 4th order (45%), with a range from 1st to 10th order (Fig 3C). Conversely, *H.* (*Feropsyche*) species has less than 20% of its records above 4th order, with a range from 1st to 8th order (Fig 3D), indicating significant differences ($F_{(1)}$=45.93, *p*-value=0.001). The species occur in areas with annual precipitation ranging from 349 to 4427 mm³/year. The *H.* (*Feropsyche*) species has 47% of its records in areas with less than 1000 mm³/year (range 349–4427 mm³/year) (Fig 4A), while *H.* (*Cochliopsyche*) species has 74% of its records in areas above 1000 mm³/year (range 959–3798 mm³/year) (Fig 4B), with significant differences ($F_{(1)}$=58.52, *p*-value=0.001). Temperature gradients range from −17°C to 40°C. The *H.* (*Feropsyche*) species is the only subgenus found in areas with negative temperatures, with a range from −17°C to 40°C (Fig 4C). In contrast, *H.* (*Cochliopsyche*) species are found between 5°C and 35°C (Fig 4D), with significant differences ($F_{(1)}$=55.10, *p*-value=0.001).

## Distribution patterns of New World

The northernmost species of *H.* (*Feropsyche*) is *H.* (*Feropsyche*) *borealis*, recorded in Canada at approximately 47°N, while the southernmost is *H.* (*Feropsyche*) *caligata*, found in Chile at around 39°S. In *H.* (*Cochliopsyche*), the northernmost species is *H.* (*Cochliopsyche*) *vazquezae* from Mexico (17°N), and the southernmost is *H.* (*Cochliopsyche*) *opalescens* from Uruguay (32°S). The highest species richness in both subgenera is concentrated between latitudes 0° and 24°S (Fig 1).

In the New World, biodiversity hotspots *H.* (*Feropsyche*) are concentrated in three main regions: (i) northwestern Colombia extending through Costa Rica, Nicaragua, and Panama, with 231 records representing 25 species; (ii) the Greater Antilles, with 165 records of 22 species; and (iii) eastern Brazil, with 93 records of 13 species (Fig 2A). For *H.* (*Cochliopsyche*), three prominent hotspots include: (i) northern Peru, much of Colombia, and Ecuador, with some extension into the Amazon basin; (ii) the Central Amazon in Brazil; and (iii) eastern Brazil, encompassing the Central Atlantic Forest Ecological Corridor and adjacent areas, and high-altitude areas of the Caatinga domain (Fig 2B).

Species of *Helicopsyche* occur across 90 freshwater ecoregions [*sensu* 66]. The Northeastern Mata Atlântica Freshwater Ecoregion harbors the highest species richness (23 spp.), followed by the South America Caribbean Freshwater Ecoregion (15 spp.), and both the Amazonas Lowlands and Paraíba do Sul Freshwater Ecoregions (14 species each). *Helicopsyche* (*Cochliopsyche*) is predominantly recorded from the Amazonas Lowlands (10 spp.) and Northeastern Mata Atlântica (8 spp.) Freshwater Ecoregions, whereas *H.* (*Feropsyche*) is more frequent in the South America Caribbean (15 spp.), San Juan (Nicaragua and Costa Rica), and Paraíba do Sul (12 species each) Freshwater Ecoregions (see S2 Table for details).

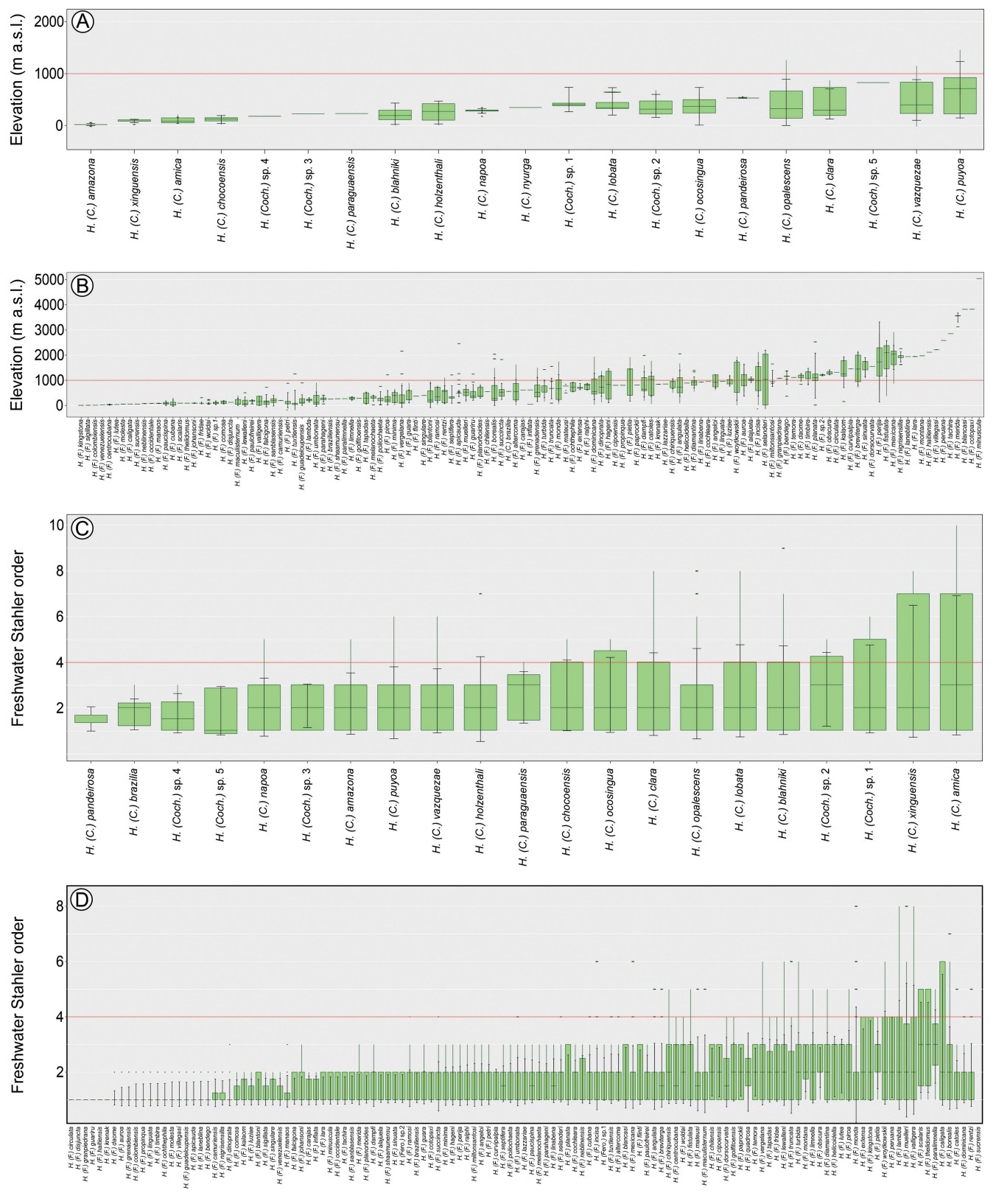

**Fig 3. Boxplot of biogeoclimatic variables of New World *Helicopsyche* species.** A. Elevation range of *Helicopsyche* (*Cochliopsyche*) species; B. Elevation range of *Helicopsyche* (*Feropsyche*) species; C. Freshwater Stahler order range of *Helicopsyche* (*Cochliopsyche*) species; D. Freshwater Stahler order range of *Helicopsyche* (*Feropsyche*) species.

## Distribution modelling

After conducting correlation testing, nine raster variables were found to be uncorrelated, belonging to five groups (bioclimatic, elevation, precipitation, solar radiation, and wind speed), as listed in Table 3 (for more details, see S3 Table and S1 Fig). Among the four algorithms tested, all yielded AUC values higher than the cut-off value (S2 Fig). These values were subsequently used to generate environmental suitability maps for the subgenera (Fig 1).

Distribution modelling indicates high environmental suitability (values > 0.7) for *H.* (*Cochliopsyche*) species across broad areas of the Neotropical region, particularly throughout the Brazilian and Chacoan subregions. Areas of notable suitability include: (i) the Amazon Basin, (ii) the Atlantic Forest dominion, and (iii) high-altitude regions in the eastern portions of the Caatinga and Cerrado dominions (Fig 1A and C). In contrast, areas of low suitability are primarily found in the (i) Mexican and South American transition zones; (ii) the Mesoamerican dominion, and (iii) the Antillean subregion. Within the Brazilian subregion, low suitability is also observed in the (iv) Pará, Xingu-Tapajos, Cerrado, Chacoan, and Pampean provinces, as well as across the entire (v) Patagonian region (Fig 1A and C).

For *H.* (*Feropsyche*) species, the model predicts high environmental suitability (> 0.7) primarily in the Nearctic region, especially: (i) in South Florida (Florida Keys), where no species records currently exist. In the Mexican transition zone, (ii) southern portions of the Chihuahuan Desert, Sierra Madre Occidental and Oriental, and the Trans-Mexican Volcanic Belt also show high suitability. In the Neotropical region, four major areas exhibit high suitability: (iii) the entire Antillean subregion; in the Brazilian subregion (iv) most part of the Mesoamerican dominion, excluding the coastal areas of the Veracruzan province and nearly all of the Yucatán Peninsula; (v) in the Pacific dominion, the Guatuso–Talamanca, Puntarenas–Chiriquí, and Chocó–Darién provinces, as well as high-altitude (Andean Cordillera) and coastal regions of Colombia and Venezuela—except much of the Guajira and Magdalena provinces; (vi) in the Boreal Brazilian dominion, the western portions of the Guiana Lowlands and Yungas provinces; and Chacoan subregion (vii) the entire Paraná domain and areas east of Caatinga Cerrado and Chacoan provinces where they are bordered by the Atlantic Forest and Paraná provinces (Fig 1D and F). Conversely, areas of low environmental suitability for *H.* (*Feropsyche*) species are primarily located in: (i) most of the South American transition zone; (ii) nearly all the Boreal Brazilian, South Brazilian, and Southeastern Amazonian dominions; and (iii) most of the Chacoan dominion, except for eastern portions of Caatinga, Cerrado and Chacoan provinces (Fig 1D and F).

## Discussion

### Biodiversity knowledge shortfalls and challenges

Our results demonstrate the clear increment of species descriptions and cataloguing of distributional records, mainly in under-explored areas of Neotropical region. This increase can be related with the development of trichopterology in the Neotropical, Nearctic and Patagonian regions [as indicated in 33,67], cooperation among researchers of NT with other researchers' groups [e.g., 24,36], and the establishment of research groups in different countries of the Neotropical and Patagonian regions [e.g., 23,38,68]. After Johanson [30,35,41,45,69], who established the modern panorama of Helicopsychidae taxonomy in the Neotropical, Nearctic and Patagonian regions, several contributions have been made by local researchers [e.g., 32,37,40,62,70], but no one with a broad perspective.

However, our results indicate more dramatic scenarios for *Helicopsyche*, with up to 75% of the fauna still unknown and numerous unexplored areas, when compared with previous estimates for Trichoptera in the NT [23,24], both with around 50% unknown species] (Table 2). These results are consistent when looking at the number of known species and the vast

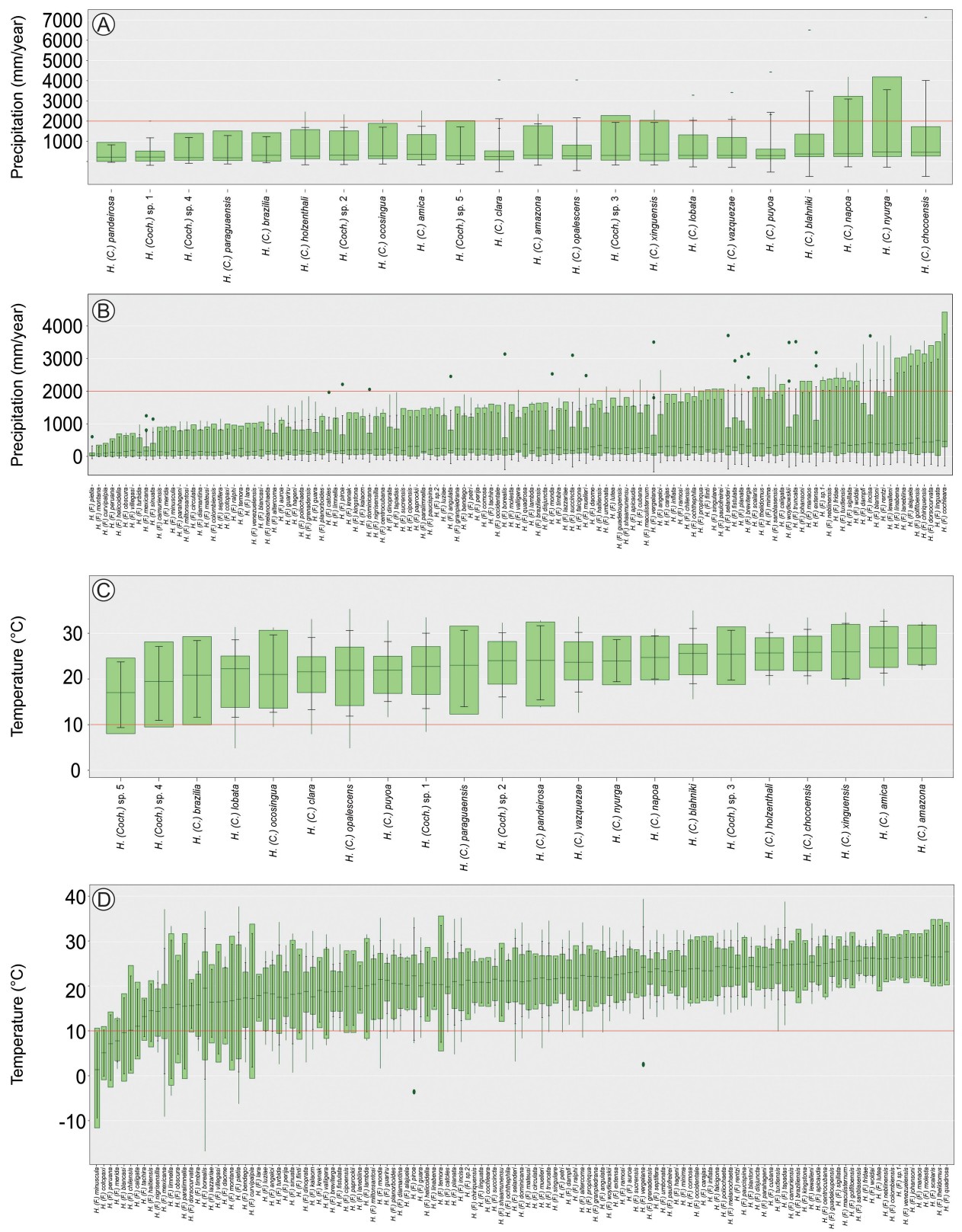

**Fig 4. Boxplot of biogeoclimatic variables of New World *Helicopsyche* species.** A. Annual mean precipitation range of *Helicopsyche* (*Cochliopsyche*) species; B. Annual mean precipitation range of *Helicopsyche* (*Feropsyche*) species; C. Annual mean temperature range of *Helicopsyche* (*Cochliopsyche*) species; D. Annual mean temperature range of *Helicopsyche* (*Feropsyche*) species.

**Table 3. Environmental variables used for species modelling, using 80% correlation cut-off.**

| Variables group | Code | Variables |
|---|---|---|
| Bioclimatic | Bio_7 | Temperature Annual Range (Annual Precipitation- Min Temperature of Coldest Month) |
| | Bio_2 | Annual Precipitation |
| Elevation | Elev | Elevation |
| Precipitation | Prec_01 and 08 | Precipitation (mm) |
| Solar radiation | Srad_03, 04 and 07 | Solar radiation (kj m-2 day-1) |
| Wind speed | Wind_01 | Wind speed (m s-1) |

areas with high suitability and little or no research effort, so efforts should be made to describe new species. In addition to unknown species, some valid species exhibit weak circumscription or inconsistencies [see 32,40]. In *H.* (*Cochliopsyche*), there is little difference in genital characteristics (the primary features for species distinction), thus, detailed description of genitalia, standardization, and better circumscription of species are relevant to avoid misidentifications.

The Wallacean shortfall is evident in the limited data available on species distribution [8]. Distributional information is often restricted to type localities or adjacent areas (Fig 2A and B), a trend that is also reflected in our results. This highlights the urgent need to increase faunistic surveys in unexplored areas with high environmental suitability (Fig 1C,F, and H) and to analyze the material already housed in collections. Key regions with significant research effort and species richness include: (i) Antillean subregion in Hispaniola, Puerto Rico – Virgin Islands, Windward & Leeward Islands, and Cuba Cayman Islands (eastern and western extremes) Freshwater Ecoregions; (ii) Mesoamerican dominion in San Juan, Estero Real – Tempisque, Chiriqui, Isthmus Caribbean, Chagres, South America Caribbean Drainages – Trinidad and south portion of Magdalena – Sinu Freshwater Ecoregions, (iii) Brazilian dominion only near Manaus city, Amazonas, and (iv) Parana dominion in Paraíba do Sul, Northeastern Mata Atlântica, São Francisco, and Fluminense Freshwater Ecoregions (Fig 1G and H).

In contrast, areas such as (i) Boreal and South Brazilian dominions; and (ii) areas east of Caatinga Cerrado and Chacoan provinces where they are bordered by the Atlantic Forest and Paraná provinces remain largely underexplored despite showing high environmental suitability, indicating strong potential for species distribution. Prioritizing these regions for future faunistic surveys targeting Helicopsychidae is essential to fill existing knowledge shortfalls. These efforts will likely lead to the discovery of previously undescribed taxa and a more comprehensive understanding of species distributions [8]. In doing so, they will help address both Linnean and Wallacean shortfalls, while also contributing to resolving Haeckelian and Hutchinsonian shortfalls if semaphoronts and abiotic conditions are thoroughly documented.

Given the Wallacean and Linnean shortfalls in caddisfly research, our understanding of species diversity and taxonomic groups remains incomplete, particularly within Trichoptera [36], where a male-biased focus persists—a gap that aligns with the Haeckelian shortfall [more details, 9]. Biodiversity biases often stem from research concentrated in easily accessible areas with prior survey efforts [3]; however, in this case, the sexual bias is significantly amplified. Therefore, acknowledge that all subsequent discussions are inherently influenced by these knowledge shortfall, reflecting the broader discrepancies in our understanding of New World Helicopsychidae, a scenario that is similarly prevalent across most insect orders.

The *H*. (*Feropsyche*) has 36 valid species, and *H*. (*Cochliopsyche*) present ten valid species with adult females deposited (in material examined), but not formally described in published studies (Table 1). Both subgenera have specimens of immature stages deposited in museums and collections that have not yet been associated with adult males (Table 1). These data demonstrate the enormous potential of the material to be described and fulfil the biodiversity shortfalls. The description of these semaphoronts can contribute to a better circumscription of the species, the correction of possible taxonomic errors, and the formation of a complementary database for studies related to the understanding of relationships and biogeographic history [9].

To our knowledge, this is one of first inference describing environmental gradients for Trichoptera species [preceded by the work of [25,26], which investigating the influence of elevation on Trichoptera]. Here are addresses Hutchinsonian deficits by empirically defining species' abiotic limits and identifying areas with a higher probability of distribution in regions of high suitability, focusing on the niches of *H*. (*Feropsyche*) and *H*. (*Cochliopsyche*). This also provides evidence of niche differentiation between the two living lineages of Helicopsychidae in the New World. Additionally, these data help identify species more vulnerable to extinction due to their narrow distribution gradients in biogeoclimatic variables. However, we acknowledge that our data are limited by current knowledge of species distribution, and future research may expand the distribution range of species.

## Distribution patterns and potential distribution in New World

The initial step in addressing the challenges posed by BKS is the compilation of inventories, checklists, descriptions of new species, and new distribution records [36]. However, it is equally important to consider other aspects, such as ecology, biology, phenology, and ethology, which can inform biodiversity conservation policies and the consequences of anthropogenic interventions (e.g., climate change, suppression of riparian vegetation) [23,25,26,71]. This study establishes the environmental gradients for the *H*. (*Feropsyche*) and *H*. (*Cochliopsyche*) species. Despite the considerable range of elevation, temperature, and precipitation observed across both subgenera, the data indicated a higher prevalence of *H*. (*Feropsyche*) species in coastal areas of tropical and temperate environments, across a wide elevation gradient (0–5042 m a.s.l.), in low order streams and in areas with high rainfall (Figs 3 and 4). On the other hand, the *H*. (*Cochliopsyche*) species are restricted to tropical environments, with prevalence in rainforests and lowland areas. Of the distributional records, 55% are in higher order streams (>4th order) (Figs 3 and 4).

The results presented here concur with previous hypotheses proposed by Monson et al. [72], Flint [62,63], and Johanson [41,45]. However, this study provides a comprehensive overview of the distribution and analyzes the differences in environmental space between the subgenera. In addition to the morphological differences, these factors contribute to the accumulation of divergence between the NW Helicopsychidae lineages. Currently, there is still no clear resolution regarding their phylogenetic relationship. Some hypotheses suggest that they are sister groups [73], while others posit that they are distant clades [30], further complicating their taxonomic placement.

Despite its wide distribution in tropical regions, *H*. (*Cochliopsyche*) is largely restricted to the type locality or adjacent localities (S1 Table). Distribution records are scarce and generally consist of single collections with a small number of specimens (S1 Table). This is primarily due to the greater concentration of collection and research efforts on Trichoptera in small, oxygenated freshwater environments, where most of the group's diversity is concentrated [34]. It is therefore recommended that research efforts be directed towards the at inventorying of South America's large rivers, with priority areas for *H*. (*Cochliopsyche*) studies (areas of high suitability and low research effort) including all South America's large river basins (e.g., Amazon, Orinoco, Paraná, and São Francisco River basins), with a particular focus on the large size rivers (Fig 1C).

The species of *H*. (*Feropsyche*) are distributed across a wide range of habitats, from the temperate areas of the Nearctic Region to the Andean areas of Chile (Fig 1E). However, the areas with the highest environmental suitability are throughout Central America (including the Antilles), the Florida Keys, areas east of the Andes between Argentina and

Bolivia (Chaco domain), and the coastal region of northwest and east South America. Much of continental Central America, the Antilles, and the northwest of South America have been well sampled (Fig 1D). However, areas of the Chaco domain, the northern portion and Central Corridor of Atlantic Forest, and high-altitude areas of the Cerrado and Caatinga domains remain poorly sampled (Fig 1E). These areas should be prioritized for research efforts, in agreement with the Brazilian caddisfly studies of Santos et al. [24], which indicated the to prioritize research efforts in the Northeastern and Central-Western regions of Brazil.

The Patagonian subregion is one of the least researched in the NT, with both groups exhibiting low suitability for this region. However, 11 species of *H.* (*Feropsyche*) are found in areas with minimum temperatures below 0°C (e.g., *H.* (*Feropsyche*) *borealis*, −17°C; *H.* (*Feropsyche*) *minuscula*, −12°C) (Fig 1D). The data available do not permit us to determine whether the low suitability is (i) due to the absence of favorable environmental conditions, or (ii) to the scarcity of data, which resulted in the model's limitation for this area. In general, there is a considerable range of distribution records for the *Helicopsyche* in the NW.

The species are distributed across a wide environmental gradient, with *H.* (*Feropsyche*) being particularly notable for their ability to occur in various freshwater environments across the New World. This includes a wide range of altitudes, temperatures, and precipitation levels, with a preference for low-order environments (Figs 1,3, and 4). In contrast, *H.* (*Cochliopsyche*) species typically occur in lowlands, environments with higher average temperatures, and a preference for large rivers, reflecting a restriction in their occurrence to large river basins (Figs 1,3, and 4). Thus, the results indicate that the two lineages have specialized in different niches: *H.* (*Feropsyche*) specialized in smaller (low-order) environments, while the *H.* (*Cochliopsyche*) specialized in larger (high-order) environments. Similar patterns are also observed in other Trichoptera lineages, including Hydrobiosidae (*Amphichorema* and *Atopsyche*), and Glossosomatidae (*Culoptila* and *Cariboptila*) [33].

The results of this study highlight critical conservation implications for Helicopsychidae species in the New World. The identification of environmental gradients and the discovery of species with restricted distribution in high-suitability and low research effort areas emphasize the need for targeted conservation efforts. Areas with high species richness and environmental suitability, such as parts of Central and South America, should be prioritized for further research and protection. The underexplored regions, especially tropical rainforests and the dry diagonal zones (Caatinga, Cerrado, and Chaco), represent significant gaps in knowledge and potential hotspots for biodiversity. Increasing faunistic surveys in these regions will not only enhance species descriptions but also help mitigate the risk of extinction for species with narrow distribution ranges, which are more susceptible to threats such as climate change and habitat loss. Moreover, the conservation of riparian habitats, particularly in large river basins, is critical to maintaining the integrity of ecosystems supporting these specialized species. By addressing both Linnean, Wallacean, Haeckelian, and Hutchinsonian and other shortfalls, future research can refine conservation strategies and promote the preservation of these important freshwater ecosystems.

## Conclusion

The results highlight recent progress in the taxonomy and distributional cataloguing of New World Helicopsychidae, particularly within underexplored regions of the Neotropics. Despite a substantial increase in described species, significant gaps in biodiversity knowledge remain. Current estimates suggest that up to 75% of Helicopsyche species in the region are still undescribed, emphasizing the urgent need for continued research efforts—ideally incorporating all semaphoronts—to fully document and understand the diversity of NW Helicopsychidae

The distribution of *Helicopsyche* species reveals complex patterns influenced by environmental factors such as elevation, temperature, and precipitation. Species of *H.* (*Cochliopsyche*) are typically associated with larger river systems in warmer climates and extensive river basins, whereas *H.* (*Feropsyche*) species show broader ecological tolerance, occurring across diverse freshwater habitats throughout the Americas, particularly in low-order streams. These contrasting patterns reflect distinct ecological specializations between the two subgenera.

Sampling efforts across the New World remain uneven, with the majority of data derived from historically surveyed regions (e.g., the Antilles) or areas with established research institutions (e.g., Central America, the Amazon, and the Atlantic Forest). These regions, due to their high species richness, should be prioritized in conservation planning, especially those identified as biodiversity hotspots. However, vast areas remain poorly explored, including the Amazon lowlands, northern and central Atlantic Forest, high-altitude environments in the Caatinga and Cerrado domains, and transitional zones between major phytogeographic domains. Targeted research in these underrepresented areas is essential.

This study represents a foundational step toward a more comprehensive database of Trichoptera diversity in the New World. It provides critical tools for delineating biogeographic patterns and identifying knowledge gaps, thereby enhancing our understanding of Helicopsychidae diversity. The compilation presented herein establishes a valuable baseline for future taxonomic, ecological, and conservation-oriented studies on this family. and may inspire future research proposals for this family or other Trichoptera taxa using similar approaches.

## Supporting information

**S1 Table. Summary of georeferenced distribution data for all New World Helicopsychidae species.**
(XLSX)

**S2 Table. Summary of distribution in Freshwater Ecoregion (*sensu* Abell et al. 2015) and limits data for biogeoclimatic variables of all New World Helicopsychidae species.**
(XLSX)

**S3 Table. Pearson correlation between environmental variables.**
(XLSX)

**S1 Fig. Boxplot with the result of Pearson correlation between environmental variables.**
(TIFF)

**S2 Fig. Boxplot with result of evaluation of distribution modelling algorithms using the area under curve (AUC) method.**
(TIF)

## Acknowledgments

We extend our sincere thanks to Dr. Neusa Hamada, Dr. Ana Maria Pês, and Dr. Gleison R. Desidério for kindly providing data and permitting the analysis of material from the Laboratório de Citotaxonomia e Insetos Aquáticos of the Instituto Nacional de Pesquisas da Amazônia (LACIA-INPA). We are also grateful to the Museum of Comparative Zoology (MCZ) at Harvard and to Dr. Crystal A. Maier for granting access to, and assisting with the analysis of, biological material from the MCZ collection. Our appreciation further extends to Dr. Ralph Holzenthal and Dr. Robin E. Thomson at the University of Minnesota, and to Dr. Dikow Torsten of the Smithsonian Institution for kindly loaning specimens from institutional collections for analysis at the MCZ.

## Author contributions

**Conceptualization:** Rafael Pereira, Adolfo Calor.

**Data curation:** Rafael Pereira.

**Formal analysis:** Rafael Pereira.

**Funding acquisition:** Adolfo Calor.

**Investigation:** Rafael Pereira.

**Methodology:** Rafael Pereira.

**Resources:** Adolfo Calor.

**Supervision:** Adolfo Calor.

**Validation:** Rafael Pereira.

**Visualization:** Rafael Pereira.

**Writing – original draft:** Rafael Pereira.

**Writing – review & editing:** Adolfo Calor.

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
