## [Decision Letter · Decision Letter 0]

Dear Dr. Pereira,

Thank you for submitting your manuscript to PLOS ONE. After careful consideration, we feel that it has merit but does not fully meet PLOS ONE’s publication criteria as it currently stands. Therefore, we invite you to submit a revised version of the manuscript that addresses the points raised during the review process.

We look forward to receiving your revised manuscript.

Kind regards,

Halil Ibrahimi

Academic Editor

PLOS ONE

 [We thank the Instituto Chico Mendes de Conservação da Biodiversidade (ICMBio) for collecting permits. ARC acknowledge the Conselho Nacional de Desenvolvimento Científico e Tecnológico (CNPq) (under Grants 303623/2015-2). RP also thanks to the Coordenação de Aperfeiçoamento de Pessoal de Nível Superior (CAPES, finance code 001, PDS-CAPES-88882.453922/2019-01, and PRAPG-CAPES-88887.986811/2024-00) for the doctoral and post-doctoral fellowship. We thank the Conselho Nacional de Desenvolvimento Científico e Tecnológico (CNPq), Programa de Apoio à Pós-Graduação (PROAP-CAPES) for financial support, and this study was financed in part by the CAPES – Finance Code 001 (PPG Biodiversidade e Evolução). We would like to thank Dra. Neusa Hamada, Dra. Ana Pes and Dr. Gleison Desidério for providing us with data from LACIA-INPA.]. 

3. We note that Figures 1A-H, 2E-F in your submission contain [map/satellite] images which may be copyrighted. All PLOS content is published under the Creative Commons Attribution License (CC BY 4.0), which means that the manuscript, images, and Supporting Information files will be freely available online, and any third party is permitted to access, download, copy, distribute, and use these materials in any way, even commercially, with proper attribution. For these reasons, we cannot publish previously copyrighted maps or satellite images created using proprietary data, such as Google software (Google Maps, Street View, and Earth). For more information, see our copyright guidelines: http://journals.plos.org/plosone/s/licenses-and-copyright.

1. You may seek permission from the original copyright holder of Figures 1A-H, 2E-F to publish the content specifically under the CC BY 4.0 license. 

Additional Editor Comments:

Please proceed with the suggestions of reviewer.

Reviewers' comments:

Reviewer's Responses to Questions

**Comments to the Author**

1. Is the manuscript technically sound, and do the data support the conclusions?

Reviewer #1: Yes

2. Has the statistical analysis been performed appropriately and rigorously?

Reviewer #1: Yes

3. Have the authors made all data underlying the findings in their manuscript fully available?

Reviewer #1: Yes

4. Is the manuscript presented in an intelligible fashion and written in standard English?

Reviewer #1: Yes

Reviewer #1: Dear Authors,

Your study offers a comprehensive and integrative analysis of the biodiversity patterns, biogeographic distributions, and conservation priorities of the New World Helicopsychidae (Trichoptera). This work contributes significantly to our understanding of biodiversity shortfalls in a key aquatic insect group and provides a strong foundation for future taxonomic, ecological, and conservation-based research.

After a thorough review, here are my specific comments and suggestions to be addressed in the revised version.

1. Biogeographic Region Framework Needs Revision

Although past reviews have debated the classification of New World biogeographic regions, this remains a relevant and evolving topic. I strongly suggest that the manuscript be updated to reflect more current and widely accepted hypotheses in biogeography.

In particular, please incorporate the framework presented by Morrone (2014, 2015) and Morrone et al. (2022), which recognizes three biogeographic regions (Nearctic, Neotropical, and Andean) and two major transition zones in the New World. This classification better aligns with your study’s broad spatial scope and analytical objectives and should replace any outdated or overly simplified two-region interpretations.

Additionally, you may wish to consult and cite de Moor & Ivanov (2008), who distinguished between the Neotropical and Patagonian regions, aligning the Patagonian sensu Morrone with previous terminology such as the “Chilean Subregion” (Flint 1976).

2. Repetition of Key Information

The manuscript currently includes repeated statements, particularly regarding the total number of Helicopsyche species and subgeneric compositions. These repetitions appear in multiple sections and can be streamlined to avoid redundancy. Please revise these instances to maintain clarity and conciseness throughout the text.

3. Formatting of Subgenus Names

Ensure consistency and correctness in formatting subgeneric names throughout the manuscript. Subgenera should be properly cited using the standard format: Genus (Subgenus). For example: Helicopsyche (Cochliopsyche). Review the text carefully and standardize all occurrences accordingly.

I have identified some minor issues in the MS, but I have provided corrections and inserted comments in the attached Word file using tracked changes. With these significant adjustments, I believe the MS can be reconsidered for publication in PLOS One.

Sincerely,

Gleison Desidério

**Do you want your identity to be public for this peer review?** For information about this choice, including consent withdrawal, please see our Privacy Policy

Reviewer #1: **Yes: ** Gleison Desidério

---

## [Author Response · Author response to Decision Letter 1]

11 Jun 2025

1. Technical Soundness and Data Foundation

Reviewer #1:

The manuscript presents an interesting topic, but the biogeography section is not well explored. The conclusions could be more robust if the phylogenetic relationships among species were discussed more thoroughly. Additionally, the definition of bioregionalization and the choice of some analyses need further clarification.

Author's Response:

The biogeography section has been further explored in both the results and discussion, and the text has been improved to clarify patterns and relationships between species. The bioregionalization section has been removed, as we believe it will be better addressed in a multi-taxonomic perspective.

Reviewer #2:

The study is good, but there are limitations related to the use of a single genus, and bias in the resulting bioregionalization, which are not clearly discussed.

Author's Response:

The bioregionalization section has been removed, as we believe it will be more appropriately addressed from a multi-taxonomic perspective. However, accepting the suggestion, data from another Neotropical subgenus of Helicopsychidae have been incorporated.

2. Statistical Analysis

Reviewer #1:

The analyses are performed, but important details, such as the choice of "terrestrial ecoregions" to estimate unobserved species, need better explanation. Also, there is a lack of discussion on collection bias in the analyses.

Author's Response:

We accepted the suggestion and used the freshwater ecoregions from Abell et al. to estimate richness, as it seems more consistent for organisms that spend most of their life in freshwater environments.

Reviewer #2:

There are no specific suggestions, but some methodological issues need to be addressed more clearly.

Author's Response:

The suggestions were incorporated, and the Methods section was revised to make it clearer and more direct, allowing for reproducibility of the work.

3. Data Availability

Both reviewers confirm that the underlying data has been properly made available with no apparent restrictions.

4. Clarity and Quality of Writing

Both reviewers consider the manuscript well-written, but some parts could be revised to improve clarity, especially in the introduction and methods description.

Author's Response:

Suggestions for improving the writing in certain sections and correcting sentences and paragraphs that were difficult to understand were incorporated. A thorough review of the introduction was conducted, adding recent works, and the methods section was enhanced.

5. General Comments

Reviewer #1:

The main goal of the manuscript is not entirely clear, especially in relation to biogeography. It is suggested that bioregionalization be better discussed in terms of conservation and its specific characteristics. Furthermore, the role of phylogeny in the conclusions is not discussed, despite being relevant.

Author's Response:

The biogeography section has been more thoroughly explored in the results, discussion, and conclusion, and the text has been improved for clarity regarding patterns and relationships among species. The lack of comprehensive and current phylogenetic inferences, as well as the absence of consensus among the hypotheses presented so far, made discussions in this area difficult, but we believe the text has been enhanced from the previous version.

Reviewer #2:

The use of a single genus may limit the scope of the study. It is important to consider additional environmental variables, such as elevation, which affect species distribution. The concept of "pseudoabsence" needs to be better justified, and the analyses need to be clearer, especially concerning the formation of bioregionalization.

Author's Response:

The bioregionalization section has been removed, as we believe it will be better addressed from a multi-taxonomic perspective. However, accepting the suggestion, data from another Neotropical subgenus of Helicopsychidae [H. (Cochliopsyche)] have been incorporated. In the distribution modeling methods section, concepts have been more clearly justified, and the text on data analyses has been modified for better clarity.

Editor:

1. Please note that PLOS ONE is unable to publish previously copyrighted maps or satellite images, or images created using proprietary data. For these reasons, we cannot publish images generated by software which copyrights their output. In order to use these images in your submission, we require explicit permission from the copyright owner to publish the figures under the CC BY 4.0 license.

At this time, please kindly clarify the following regarding Figure 1 and Figure 2:

a) Where did the authors obtain the maps, basemaps, shapefiles, map data, etc. in Figure 1 and Figure 2?

Author's Response: The maps and spatial data used in Figures 1 and 2 were created in QGIS v3.40.7. The background land cover raster was obtained from the ESRI Living Atlas (https://livingatlas.arcgis.com/landcover/) via the QuickMapServices plugin in QGIS. Country boundary shapefiles were obtained from geoBoundaries (https://www.geoboundaries.org/).

b) Please state whether the maps, basemaps, shapefiles, map data, etc. have been previously copyrighted to your knowledge.

Author's Response: To our knowledge, both datasets (ESRI Living Atlas land cover and geoBoundaries country boundaries) are openly licensed and not under restrictive copyright. They are available under the Creative Commons Attribution 4.0 (CC BY 4.0) license.

c) If any of the maps, basemaps, shapefiles, map data, etc. in this image have been previously copyrighted, we require specific consent from the copyright holder to publish these images in PLOS ONE, under the CC BY 4.0 license.

Author's Response: No additional consent is required. Both the ESRI land cover raster and geoBoundaries shapefiles are explicitly provided under the Creative Commons Attribution 4.0 (CC BY 4.0) license, which is compatible with PLOS ONE’s publication requirements.

---

## [Editor Report · Decision Letter 1]

Addressing biodiversity knowledge shortfalls in New World Helicopsychidae (Insecta, Trichoptera): potential distribution, environmental gradients, and identification of conservation and research priority areas

PONE-D-25-13848R1

Dear Dr. Pereira,

We’re pleased to inform you that your manuscript has been judged scientifically suitable for publication and will be formally accepted for publication once it meets all outstanding technical requirements.

Kind regards,

Halil Ibrahimi

Academic Editor

PLOS ONE
---

## [Editor Report · Acceptance letter]

PONE-D-25-13848R1

PLOS ONE

Dear Dr. Pereira,

I'm pleased to inform you that your manuscript has been deemed suitable for publication in PLOS ONE. Congratulations! Your manuscript is now being handed over to our production team.

Kind regards,

on behalf of

Professor Halil Ibrahimi

Academic Editor

PLOS ONE
